# Prevalence and Factors Related to *Leishmania infantum* Infection in Healthy Horses (*Equus caballus*) from Eastern Spain

**DOI:** 10.3390/ani13182889

**Published:** 2023-09-11

**Authors:** Lola Martínez-Sáez, Quentin Dulac, Esperanza Montaner-Angoiti, Pablo Jesús Marín-García, Lola Llobat

**Affiliations:** 1Molecular Mechanisms of Zoonotic Diseases (MMOPS) Research Group, Facultad de Veterinaria, Universidad Cardenal Herrera-CEU, CEU Universities, 46115 Valencia, Spain; lola.martinez@alumnos.uchceu.es (L.M.-S.); quentin.dulac@alumnos.uchceu.es (Q.D.); esperanza.montaner@uchceu.es (E.M.-A.); 2Department of Animal Production and Health, Veterinary Public Health and Food Science and Technology (PASASPTA), Facultad de Veterinaria, Universidad Cardenal Herrera-CEU, CEU Universities, 46115 Valencia, Spain; pablo.maringarcia@uchceu.es

**Keywords:** equine, *Leishmania*, One Health, parasitic infection, zoonosis

## Abstract

**Simple Summary:**

Leishmaniosis is a zoonotic disease transmitted by sandflies. The main reservoir is the dog, although the number of species involved is increasing. Horses, due to their high level of contact with humans and ability to control the disease, could be a silent reservoir. However, data related to the prevalence of *Leishmania* spp. infection in horses are scarce, especially in Europe. In this study, the prevalence and factors related *to L. infantum* infection in apparently healthy horses have been studied. The results indicate that the equine prevalence is elevated, and the main factors related to infection were equine breed, morphotype, outdoor living, use, and season. Horses with a calm temperament and outdoor living conditions have a higher prevalence of infection, and the number of positive animals increases in spring, when the mean of temperature is higher. These results indicate that horses could be a silent reservoir of the parasite and that the increase in temperature due to climate change will probably increase the *Leishmania* spp. infections in all species, including humans, in the future. From a One Health perspective, to control this zoonosis it would be advisable to also incorporate measures in horses, such as the use of repellents.

**Abstract:**

Leishmaniosis is a zoonosis caused by *Leishmania* spp., an intracellular protozoan parasite. This parasite is transmitted by sandflies, and the disease is endemic in the Mediterranean basin. In recent years, the number of species which could be a reservoir of the parasite is increased. One of the most relevant species is the horse, due to their contact with humans and ability to control the disease, thus being a possible silent reservoir. In this study, we have analyzed the prevalence and factors related to *L. infantum* infection in healthy horses in the Mediterranean region. Epidemiological data and serum samples were obtained from 167 apparently healthy horses, and the presence of *L. infantum* was evaluated via the ELISA method and real-time PCR. The results show 27.5% of prevalence and that the main factors related to infection are equine breed, morphotype, outdoor living, use, and season. In conclusion, the prevalence of *L. infantum* infection in apparently healthy horses from eastern Spain (Mediterranean basin) is elevated. To control this zoonosis, it would be advisable to carry out more studies on this and other species that could be silent reservoirs of the parasite, as well as carry out measures such as the use of repellents on a regular basis.

## 1. Introduction

Leishmaniosis is a parasitic disease caused by *Leishmania* spp., which includes *L. infantum*, the most common causal agent of leishmaniosis in the Mediterranean basin. This protozoan parasite is transmitted by the phlebotomine sandflies from the Psychodidae family, the genus *Phlebotomus* being its principal vector in Europe [1,2,3]. Even though the domestic dog (*Canis lupus familiaris*) has always been considered the main reservoir of this parasite [4,5], *L. infantum* infection has been reported in different species in Europe, including cats [6], wild carnivores [7,8], wild rabbits [9,10], birds [11], reptiles [12], and horses [13]. The relevance of horses in parasite transmission is related to the close relationship between this species and humans, since they often coexist, and in addition, these animals are usually stabled near urban centers. The majority of studies about this species are conducted in New World [14], still the leishmaniosis is considered endemic in Mediterranean basin. The studies carried out in Europe indicate that the number of clinical cases of equine leishmaniosis (EL) reported has been low, and clinical manifestations are mild without visceral involvement, and cutaneous lesions tend to self-recover without treatment [15,16,17]. *L. infantum* has been identified as the etiological agent of EL with cutaneous presentation in Germany [15] and Switzerland [18], as well as in countries where this zoonosis is considered endemic, such as Italy [19], Spain [17], and Portugal [16]. However, most data on EL have been recorded from South and Central America, the main etiological agent being *L. braziliensis* [20,21,22] or *L. infantum* [23,24,25]. 

Subclinical infections of *Leishmania* spp. in healthy horses have been reported around the world, and the prevalence in endemic areas depends on the geographic areas in which the study was carried out. In Venezuela and Brazil, the prevalence of *L. braziliensis* infection in healthy horses ranged between 7.1 and 28% [26,27], whereas the prevalence of *L. infantum* in Europe seems to be lower [28,29,30,31]. Given that both the techniques used and the number of individuals evaluated differ greatly between studies, the observed prevalence varies between countries, both in the New and the Old World. In the latter, the role of horses as a reservoir of the parasite and the factors related to it have not been well studied, but undoubtedly, horses can play a very important role [32], not only as a source of food for sandflies [25], but also because their feces are a source of food for development of their larvae [33]. For this reason, it is necessary to know the prevalence of *Leishmania* infection in endemic areas, as well as the factors that influence a higher prevalence.

The aim of this work is to analyze the prevalence of *Leishmania* infection and related factors in apparently healthy horses in eastern Spain, Mediterranean basin, an endemic region.

## 2. Materials and Methods

### 2.1. Animals and Epidemiological Data

A total of 167 horses living in Valencia Community (eastern Spain, Mediterranean region) were studied, and samples were recovered from December 2022 to June 2023. None of animals included presented clinical signs compatible with *Leishmania* infection, i.e., skin lesions on the appearance of papules or nodules.

For all animals, the following epidemiological data were collected: sex (two categories: male or female), reproductive status in males (two categories: castrated or not), age (four categories: foal, less than five years old; young, between five and twelve years old; adult, between thirteen and twenty-one years old; elder, more than twenty-one years old), breed (seventeen categories), crossbreed or purebred (two categories), use (six categories: teaching, breeding, dressage, hitch, walking, and leap), type of housing (two categories: outdoor or indoor), and living or not with dogs. Breeds were classified based on their morphological types [34] as mesomorphic, meso-brachymorphic, meso-dolichomorphic, or dolichomorphic, except ponies. Samples were recovered in two periods of the year: winter (December 2022–January 2023) and spring (May 2023–June 2023). The mean temperature was annotated.

### 2.2. Sample Collection, Serological Analysis, and DNA Extraction

In order to detect circulating anti-*Leishmania* antibodies, as well as parasite DNA in the animals included, ten milliliters of whole blood was taken via jugular venipuncture using Vacutainer tubes with or without anticoagulant. Samples without anticoagulant were maintained at room temperature to obtain serum aliquots, which were stored at −80 °C until processing. Serological testing for *Leishmania* spp. detection of specific antibodies was performed using Enzyme-linked immunosorbent assay (ELISA) test for anti-*Leishmania*-specific immunoglobulin G (IgG) antibodies (*Leishmania* vet ELISA^®^, Demeditec Diagnostics GmbH, Bonn, Germany), following the manufacturer’s instructions. The samples were analyzed per triplicate, and animals with ELISA titer > cut-off was considered seropositive. Whole blood samples recovered with EDTA anticoagulant were used for DNA extraction 24 h before recovery. Total DNA was isolated using Thermo Fisher Scientific DNA purification Kit following the manufacturer’s protocol (Thermo Fisher Scientific, Waltham, MA, USA). DNA was quantified using a Nanodrop spectrophotometer (Thermo Fisher Scientific), and only samples with A260/A280 > 1.8 were used. DNA samples were stored at −80 °C until use for PCR.

### 2.3. Real-Time PCR Analysis

The primers and probes for *L. infantum* DNA detection were chosen in the constant region of the kinetoplast DNA minicircle, and fluorogenic probes were synthesized by FAM reported molecule attached to the 5′ end, a TAMRA quencher linked to the 3′ end, and ROX as the internal passive reference dye. Primers and probe sequence are 5′-GGCGTTCTGCGAAAACCG-3′ (forward), 5′-AAAATGGCATTTTCGGGCC-3′ (reverse), and 5′-AAAATGGCATTTTCGGGCC-3′ (probe) [35]. Real-time PCR was performed in 20 µL of total volume of reaction, including 10 µL of qLUMEN MasterMix (Gquence^®^, Labbox Labware, SL, Barcelona, Spain), 300 nM of each primer, 250 nM of the fluorogenic probe, and 50–100 ng of DNA. Reactions were run in triplicate for all samples and were performed on the QuantStudio 5 Real-Time PCR System (Thermo Fisher Scientific, Waltham, MA, USA). The thermal cycle conditions consisted of a 2 min initial incubation at 50 °C, followed by 10 min at 95 °C, and 40 cycles at 95 °C for 15 s, and 60 °C for 1 min each. On each plate, a negative control (free RNA and DNA water) and positive control was included. The positive control used was a DNA sample extracted in the same way as the samples analyzed that came from a dog positive for *Leishmania* infection, confirmed by the external laboratory using the IFAT and PCR technique.

### 2.4. Statistical Analysis

Prevalence of parasitic infection was analyzed using univariable general lineal models (GLMs) with a binomial distribution and logit link function with the GENMOD procedure of the statistical program SAS (North Carolina State University, Cary, CA, USA) to explore the relation between seroprevalence and presence of *L. infantum* DNA and associated factors. Binary (for sex, reproductive status in males, crossbreed or purebred, type of housing, living or not with dogs, and period of the year) or logistic regression model (for age, breed, and use) were used using each factor as a fixed effect in their respective statistical analysis. The estimated means were then compared through pairwise comparison, and statistical significance was set at *p*-value < 0.05. 

## 3. Results

Of the 167 samples evaluated per triplicate, 27.5% were positive in the ELISA test (46/167). The presence of L. infantum DNA was confirmed in 40.7% of samples analyzed via real-time PCR (68/167). All the samples resulting positive via ELISA were confirmed through real-time PCR method, so the total prevalence of L. infantum infection in animals evaluated was 27.5% via the two methods (Table 1). 

In terms of sex, 65.3% of animals evaluated were males (109/167), 66.1% of them being castrated (72/109). In relation to age, 21% of animals were foals (less than five years), 40% were young animals (between five and twelve years old), 80% were adults (between thirteen and twenty-one years old), and 26% were old horses (more than twenty-one years old). Most horses were purebred, including a total of eighteen different breeds, and 38.3% were crossbreed animals. In terms of use, 44% of horses were used for dressage (74/167), whereas 19.2% (32/167) and 16.2% (27/167) were used for walking and teaching, respectively. Regarding type of housing, a total of 112 horses (67.1%) had outside access, and 159 (95.2%) were living with dogs. A total of 167 samples were collected from December 2022 to January 2023 (63.5%) (Table 2). 

A total of 46 (27.5%) of samples analyzed were positive for anti-leishmania antibodies and confirmed via real-time PCR. Regarding the factors evaluated, sex or castration in males, age, pure- or crossbred, and living of dogs did not have an effect on the *L. infantum* infection, whereas horse breed, life condition, use, period of sample recovery and location had an effect. Breeds with the greatest prevalence were Pony, Spanish Sport Horse (CDE), and Purebred Spanish Horse (PRE), with 75%, 57.1%, and 32.9% prevalence, respectively. Only one Haflinger and one Percheron were included in this study, and both were positive, and one of the two Irish cob horses was positive. More relevant are the results according to morphotype, where the meso-dolichomorphic animals have a higher prevalence than others (Table 3).

Outdoor horses present a higher prevalence than indoor (38.9% and 5.5%, respectively, *p* < 0.0001) (Figure 1a). The prevalence in horses used for teaching was higher than other uses, the prevalence being 88.9% (*p* < 0.0001) (Figure 1b). The number of positive samples recovered in spring (60.7%) was higher than in winter (8.5%) (*p* < 0.0001) (Figure 1c). The mean temperature was 13.1 °C and 24.1 °C, and the average rainfall was 60 and 40 mm for winter and spring, respectively [36]. Figure 1c shows the prevalence of positive animals according to the period of sample recovery and the mean temperature for each location and period. 

## 4. Discussion

This study shows the prevalence and factors related to infection of *L. infantum* in horses from eastern Spain, where this parasite is endemic. The results show a total prevalence of 27.5% via two methods (ELISA and real-time PCR) and 40.7% via real-time PCR, with the factors having an effect being horse breed, if the animal had access to outside, horse use, and the period of sample recovery. A higher number of positive animals with real-time PCR compared to ELISA can be explained through technical aspects and, mainly, on the basis of the two techniques. Whereas ELISA is an indirect technique of diagnosis, real-time PCR is a direct technique to detect parasitic DNA. In fact, other authors have observed these discrepancies in apparently healthy animals of different species including horses [37], dogs [38], cats [39], and wild animals [40]. These differences observed between positive healthy animals via serological and molecular techniques could be explained by considering the timeframe required to seroconvert, which is different between species [40]. In the case of horses, seroconversion can occur up to 13 months of age [41], and the subclinical infection is characterized by a lack of specific antibodies or by the presence of very low serum titers [15,17,42,43]. The results observed in our study support the theory that horses could be resistant hosts against infection by *Leishmania* spp., since they activate the immune response of developed T cells, and this is effective in controlling the infection [44].

This study has been realized in eastern Spain, concretely the Valencian region, where human leishmaniasis has been endemic since 1982 [45]. Although the prevalence of the disease in humans is unknown in this region, its prevalence in asymptomatic human has been reported to be around 7.9% in the south of Spain [46]. In this region, the climate and vegetation promote the dispersion of *Leishmania* vectors [47]. The presence of sand fly vectors is considered the main risk for leishmaniasis. In Spain, recent studies have reported six species of sand flies, and the most prevalent are *Sergentomyia minuta* and *Phlebotomus perniciosus* [48]. The authors analyzed the presence of L. infantum in sand flies recovered and concluded that the *P. perniciousus* are more related to *Leishmania* transmission than *S. minuta.*

Dog is consider the most common reservoir, and its seroprevalence in dogs has been studied, and in Spain, this depends on province, being between 57.1% for Balearic Islands (Mediterranean basin) and 0% for Vizcaya (north Spain) [49,50]. In the region where this study was carried out, canine leishmaniosis has a prevalence of more than 100 dogs out of 1000 [51]. More relevant is the number of asymptomatic dogs which are seropositive, which has been estimated to be around 93.5% [52]. Inasmuch as other species could also serve as reservoirs, knowing the prevalence in these other species is very important to control this zoonosis. Concretely, horses have relevance, given that they are a species in direct contact with human and usually live in urban areas or close to them. 

Data on seroprevalence in horses are limited. The only study carried out in Spain shows a prevalence of 14.3% via ELISA, carried out in Catalonia (northeast Spain) [43], and similar results have been observed in Italy [13,25]. However, in other Mediterranean countries, seroprevalence seems to be higher or lower. For example, studies realized via serological methods indicate a prevalence of 0.3% in Greece [30], 1.4% in Israel [31], and 4% in North Portugal [29], whereas Nardoni et al. (2019) observed that 36.7% of donkeys evaluated positive [53]. In South America, the seroprevalence of *Leishmania* spp. in healthy horses has been estimated to be around 27% [54], even reaching 67.3% of horses from urban areas via real-time PCR [37]. These data are in agreement with those observed in our study, confirming the sensitivity of the real-time PCR technique for the detection of *Leishmania* spp., especially in asymptomatic animals [55]. The evaluation of healthy horses in endemic areas of leishmaniasis is important, mostly in Europe, where equine leishmaniosis is produced by *L. infantum*, and its clinical presentation is usually mild, appearing as skin lesions that usually remit spontaneously, so the infection goes unnoticed in most cases [13].

Different factors have been related to *Leishmania* infection in several species of hosts. For example, one of the most relevant factors is canine breed [50]. The impact of sex is controversial, so while some authors find a higher prevalence in males than females [38,56], other find no differences [50]. With respect to age, several authors found a relationship between the prevalence of *Leishmania* infection and age in dogs. One of them reported a correlation between early age and high prevalence [38,56], whereas others found a lower prevalence in adult than younger and older [57]. In horses, the studies are limited, but it seems that breed, age, or lifestyle could be related to *Leishmania* infection.

No effect due to sex, pure- or crossbred, and living with dogs has been observed, in accordance with other studies carried out on healthy horses and dogs [13,50,54]. With respect to age, the results of other studies are contradictory. Our results indicate that age has no effect, in accordance with [13], whereas [54] show an increase in prevalence relating to age. In other species, such as dogs, seropositivity has been related to age, so in this species, the chronicity of the infection has an accumulative effect [58]. However, the immune response in horses is effective in controlling the infection [43], so there would not be this accumulative effect, unlike what happens in dogs. In terms of breed, equine breeds with a higher prevalence were CDE and PRE. In this regard, our study presents few data on certain equine breeds, so we must discuss this information carefully. However, when analyzing the results of the infection according to the morphotype, our results are in agreement with other studies, which show that meso-dolichomorphic horses present a higher prevalence than breeds with other morphotypes [13]. Meso-dolichomorphic horses, such as CDE and PRE breeds, have calm temperaments, and they are often used for non-competitive riding and recreational purposes, such as trekking or riding courses for children (teaching use). They are housing mixed and kept outside for longer than competition horses, which could make it easier for them to serve as food for the sandflies and, therefore, increase the risk of *Leishmania* spp. infection. A high prevalence of *Leishmania* infection in horses used for recreational purposes had already been observed by other authors [13,29]. This also happens in outdoor horses, where outdoor living increases the risk of infection, a factor that has been associated with this risk in other species, such as dogs [58,59]. 

The prevalence of *L. infantum* infection in spring was much higher than in winter, when the average temperature recorded was lower (24.1 °C in spring versus 13.1 °C in winter). Warmer temperatures are favorable for the development of both sand flies and parasites [60]. Tiwary et al. (2013) found the highest rate of infection in sand flies in the winter season and the lowest infection rate in the rainy season. The flushing effect of rainfall on immature sand flies could explain these results [61]. Our study was carried out in eastern Spain, where the highest rainfall occurs in autumn (average rainfall of 60 mm with relative humidity of 70%) [36]. Although there are no data on this region, this could indicate that, in this region, the lowest infectivity via the vectors is in autumn. If we consider that a time interval is necessary between the bite and its dissemination in the host organism, this lower infectivity in the rainy season could explain the lower rate of infected animals found in winter. In addition, the ability of horses to control the infection is well known [43], so this interval between the sandfly bite and the increase in the parasite load in the host, sufficient to be detected via serological and/or molecular methods, could be higher than in other species. With respect to temperature, the increase in temperature has been demonstrated as a relevant factor in *Leishmania* spp. dissemination, facilitating the presence of the parasite and its vectors [62,63,64]. These data are more relevant considering that climate change will increase temperatures throughout the year and throughout the word. Some researchers are already carrying out studies with predictive models in this regard. Concretely, [65] estimates an expansion of *L. tropica* and its vector using Ecological Niche Modeling. This modelling could be extrapolated to the expansion of *L. infantum*, as well as to its vector. In fact, *L. infantum* has been detected not only in sandflies but also in dogs and horses of cold European countries [15,66,67], and scientists already recommend including cutaneous leishmaniasis in the differential diagnosis in human patients [68].

## 5. Conclusions

In this study, a high prevalence of *L. infantum* infection has been detected in healthy horses of eastern Spain. The main related factors that increase the risk of infection are equine breed morphotype, outdoor living, and teaching use. The prevalence increases in the warm seasons, confirming that climate change and the consequent increases in temperature suggest an increase in the spread of the parasite in all species. In horses, the high prevalence observed in this study shows that horses could be an important reservoir for this zoonotic parasite, so it would be interesting to implement control measures, such as the use of repellents, mainly in horses living in urban areas. To control this zoonosis in the Mediterranean region from a One Health point of view, these control measures must be applied to this species, due to its ability to act as a silent reservoir of the parasite.

## Figures and Tables

**Figure 1 animals-13-02889-f001:**
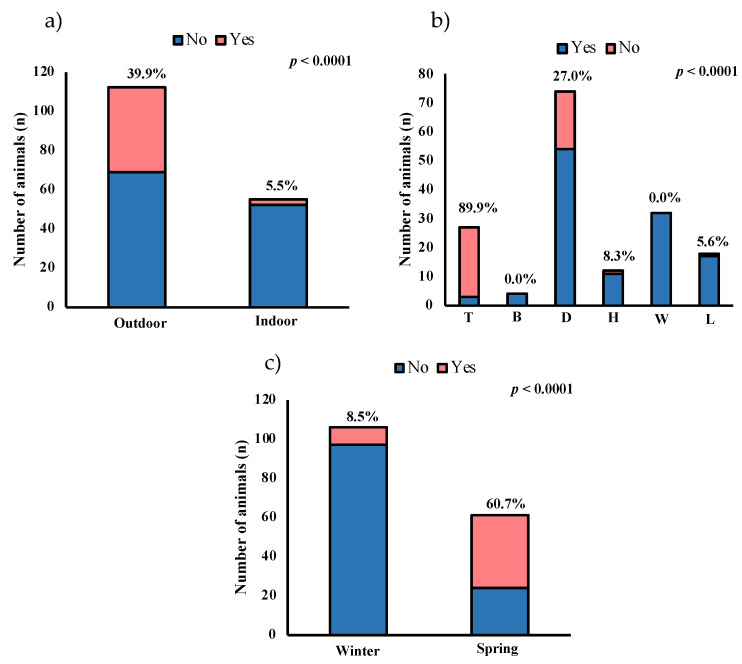
Number and percentage of positive horses via ELISA and real-time PCR regarding the following: (**a**) type of housing (outdoor or indoor); (**b**) use (T: teaching; B: breeding; D: dressage; H: hitch; W: walking; L: leap), and (**c**) season (winter, with mean temperature: 13.1 °C; spring, with mean temperature: 24.1 °C).

**Table 1 animals-13-02889-t001:** Number of positive animals with ELISA, PCR and two methods.

Method of Detection	Number of Positive Animals (%)
ELISA	46 (27.5%)
PCR	68 (40.7%)
ELISA + PCR	46 (27.5%)

**Table 2 animals-13-02889-t002:** Epidemiological data of horses included in this study. KWPN: Koninklijk Warmbloed Paard Nederland; PRE: Purebred Spanish Horse.

Variable	Categories		Number of Horses (%)
Gender	Male	Castrated	72 (66.1%)
		Not castrated	37 (33.9%)
	Female		58 (34.7%)
Age	Foal (<5 years)		21 (12.6%)
	Young (5–12 years)		40 (24.0%)
	Adult (13–21)		80 (47.9%)
	Elder (>21 years)		26 (9.6%)
Breed	Purebred (125)	Anglo-Arabian	1 (0.6%)
		Belgian Warmblood	2 (1.2%)
		Spanish Sport Horse	7 (4.2%)
		Connemara	1 (0.6%)
		Haflinger	1 (0.6%)
		Hannoverian	4 (2.4%)
		Hispano-Breton	8 (4.8%)
		Hispanic-Arabic	6 (3.6%)
		Holsteiner	1 (0.6%)
		Gypsy Cob	2 (1.2%)
		Jaca navarra	2 (1.2%)
		KWPN	5 (3.0%)
		PRE	74 (44.3%)
		Percheron	1 (0.6%)
		Pony	4 (2.4%)
		Frech Saddle Horse	5 (3.0%)
		Arabian	1 (0.6%)
	Crossbred		42 (25.1%)
Use	Teaching		27 (16.2%)
	Breeding		4 (2.4%)
	Dressage		74 (44.3%)
	Hitch		12 (7.2%)
	Walking		32 (19.2%)
	Leap		18 (10.8%)
Type of housing	Outdoor		112 (67.1%)
	Indoor		55 (32.9%)
Living with dogs	Yes		159 (95.2%)
	No		8 (4.8%)
Period of the year	Winter		106 (63.5%)
	Spring		61 (36.5%)

**Table 3 animals-13-02889-t003:** Prevalence of *L. infantum* infection according to morphotype and breed in purebred horses. KWPN: Koninklijk Warmbloed Paard Nederland; PRE: Purebred Spanish Horse; SSH: Spanish Sport Horse.

Morphotype	Equine Breed	Number of Positive Animals (%)	
Dolichomorphic	Anglo-Arabian	0	0
	Frech Saddle Horse	0	
	Holsteiner	0	
Mesomorphic	Haflinger	1	2 (5.7%)
	Arabian	0	
	Hispano-Breton	0	
	Hispanic-Arabic	0	
	Percheron	1	
Meso-brachymorphic	Gypsy Cob	1	1 (2.9%)
Meso-dolichomorphic	KWPN	0	29 (82.9%)
	PRE	25	
	Hannoverian	0	
	Belgian Warmblood	0	
	SSH	4	
Pony		3	3 (8.6%)

## Data Availability

The dataset of the current study is available from the corresponding author upon reasonable request.

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
