# Peer review of "Prevalence and Factors Related to Leishmania infantum Infection in Healthy Horses (Equus caballus) from Eastern Spain"

_animals, 2023, doi:10.3390/ani13182889_

Round 1

Reviewer 1 Report

The study "Prevalence and factors related to Leishmania infantum infection in healthy horses (Equus caballus) from Eastern Spain" aims to investigate the prevalence of Leishmania infantum infection in a sample of 167 asymptomatic horses residing in the Valencia region, Eastern Spain. The authors analyze serum and blood samples using two techniques in parallel (ELISA and qPCR) and evaluate the relationship between the positive status and a number of biological or management-related characteristics of the animals.

 General comment: Overall, the study is interesting and provides a significant contribution to the current state of knowledge on this topic, although it may not be particularly original in its approach and objectives. Certainly, the use of the two combined analytical approaches strengthens the significance of the positive data. The main concern pertains to the small number of subjects from different breeds included in the study, with the only exceptions being CPE and PRE breeds. Therefore, it is not possible to reliably compare the prevalence data among different breeds. In contrast, the Authors attribute higher sensitivity to infection in CPE and PRE breeds (e.g., line 206), which should be verified through comparable sampling. Moreover, the data in Table 2 paradoxically report higher prevalence levels, even up to 100%, for some underrepresented breeds. It is thus necessary to process the positive data differently based on the breed, for instance, by grouping underrepresented breeds into a single category, and the Authors should formulate their comments differently (e.g. Discussion, Lines 205-206), taking into account the significant limitation associated with the unequal representation of breeds.

Specific comments:

Section 2.3, Line 115: Please provide more details about the positive control, including its nature, origin, and the quantity used in the analyses.

Table 1 (section: Breed) and Table 2: To be reworked based on what is mentioned in the general comment.

Section 3, Lines 143-144: Please verify repetitions compared to what is already reported in Lines 126-129.

Discussion, Line 185: Reference 37 does not seem pertinent as it is a study conducted in dogs, not in horses.

Discussion, Lines 213-214: Although lower than the spring data, the positivity during the winter season should be discussed considering the seasonality of potential vectors in the area where the study was conducted. For example, could the detection of Leishmania DNA circulating in the blood through qPCR suggest a very recent infection acquired during the winter, outside the typical transmission season?

Check for grammatical and spelling mistakes throughout the manuscript.

Author Response

Dear reviewer,

Thank you very much for your recommendations and comments. Then we answer your questions one by one in the attach file.

Reviewer 2 Report

The authors tried to determine the prevalence of anti-Leishmania antibodies and L. infantum-specific DNA in horses from the eastern Mediterranean region. The study is very important because it is undoubtedly a severe problem in the Mediterranean region, particularly in the southern Mediterranean and North African countries. The authors claim to study factors of the diseases, which makes the study interesting for the readers. However, after reading the manuscript, I found serious shortcomings, particularly with regards to the ground on which the sampling frame was constructed and the samples selected. More epidemiological details are needed, as well as an emphasis on the reasons why this area was selected and why these clinically healthy horses were recruited for sampling? In addition, I found that the manuscript lacked the result description of the binary and logistic regression for which the factors should actually be associated with the disease. Since these are insufficient results described in the manuscript to meet the authors' claim, I will recommend rejection of this manuscript for publication unless the authors address all of these deficiencies, present the results and rewrite discussion based on the true and valid statistical analysis. Some specific comments:

Section 2.1: Please describe concretely some existing leishmaniosis data in the sampling region as well as the situation of vector prevalence. Are there only sand flies in the region or also other vectors? How about clinically ill horses? How did you choose the factors to include in the study?

Lines 126-128: Please replace “qPCR” with “real-time PCR” as I am assuming you did not quantify the DNA but rather used this PCR as a tool to screen the detectable amount of Leishmania DNA in the samples. Please correct in the entire manuscript.

Author Response

(The authors gave the same response as above.)

Reviewer 3 Report

Paragraph materials and methods: Please explain why do you chose serum and whole blood as materials to be analyzed.

Results paragraph:

-  From line 126 to line 129 have to be rewritten also because the distribution of positives between serological tests and PCR tests is not clear. The authors say that the totality of positives is 27.5%, but how is it possible if you declare at the same time the positives with PCR are 40.7%.

- I propose to the authors to make an analytical table of the positives obtained from the two methods and to insert the percentage here.

- In line 137 the authors say that 106 samples have been collected, but aren't they 167 ?

- In figure 1 I ask you to use brighter colors for the columns of the histograms in order to also improve the references called "No" and "Yes".

Discussion paragraph:

-From line 172 to line 173, specify better why the authors speak again of the 27.5% total but then report the single figure of 40.7% for the PCR test

Author Response

(The authors gave the same response as above.)

Round 2

Reviewer 2 Report

The manuscript has been revised sufficiently.

Reviewer 3 Report

The work for me can be published as the authors have made the corrections I have requested.